# Causes of Delays during Housing Adaptation for Healthy Aging in the UK

**DOI:** 10.3390/ijerph16020192

**Published:** 2019-01-11

**Authors:** Wusi Zhou, Adekunle Sabitu Oyegoke, Ming Sun

**Affiliations:** 1School of Public Administration, Hangzhou Normal University, Hangzhou 311121, China; 2School of Built Environment & Engineering, Leeds Beckett University, Leeds LS1 3HE, UK; A.Oyegoke@leedsbeckett.ac.uk; 3School of Architecture, Design and the Built Environment, Nottingham Trent University, Nottingham NG1 4FQ, UK; ming.sun@ntu.ac.uk

**Keywords:** housing adaptation, process stage, aging in place, waiting time, delays

## Abstract

Housing adaptation is a rehabilitation intervention that removes environmental barriers to help older people accommodate changing needs and age in place. In the UK, funding application for home adaptations to local authorities is subject to several procedural steps, including referral, allocation, assessment, funding and installation. The five stages need to complete in a sequential manner, often cause long delays. This study aims to investigate the timelines across these key stages of the adaptation process and examine the main causes of delays in current practice. A mixed-methods research strategy was employed. A questionnaire survey was first undertaken with all 378 local authorities in England, Scotland and Wales; it was followed by 5 semi-structured interviews and 1 focus group meeting with selected service providers, and 2 case studies of service users. The results showed that the average length of time taken to complete the whole process is relatively long, with the longest waiting time being observed at the funding decision stage. Delays were found in each of the key stages. Main causes of delay include insufficient resources, lack of joint work, legal requirements, shortage of competent contractors and the client’s decisions. These issues need to be addressed in order to improve the efficiency and effectiveness of future housing adaptation practice.

## 1. Introduction

The UK’s population is aging. For many older people, the aging process results in gradual loss in physical capacity; environmental barriers frequently turn their home into a place of embarrassment or confinement [1,2]. On the other hand, over 85% of older people have a strong desire to “stay put” in their own houses and to remain engaged in the community [3,4]. Behind this desire lies a strong attachment to the home, which keeps people busy and active, shields privacy and freedom, and boosts sense of identity and self-esteem [5,6]. Housing adaptation is recognised as an effective intervention to enhance home accessibility and to meet the changing needs of older people [7,8,9]. When health deteriorates and mobility reduces, older people can remove obstacles by adapting their houses to manage daily activities at home and participate in social life [10,11]. Both physical activity and social participation have important implications for healthy aging [12].

Housing adaptation has been defined in a variety of ways, such as a temporary rearrangement of furniture or fittings [13,14], an alteration to permanent features of the physical environment [15,16], and a physical change to the home environment including the provision of equipment [17]. This study defines housing adaptation as modification of physical features in the indoor and immediate outdoor environment (e.g., changes to the layout or the structure features and installation of fixtures and fittings) to reduce environmental barriers and restore independent living. In essence, the home environment is adapted to meet the specific needs of individuals who are experiencing difficulties in performing daily activities at home [18]. The underlying assumption that adapting the environment can improve functional performance is based on the ecological theory of aging [19,20,21]. Specifically, there is an optimal person-environment fit when personal competence is compatible with the environmental demand, while a misfit occurs when the environmental press exceeds individual ability [22,23].

In the UK, local councils have a statutory duty to provide financial assistance for housing adaptations that are assessed to be necessary to meet the special needs of disabled people and help them maintain independent living at home [24,25]. There are various funding avenues that people could access to assist with adaptations, depending on the types of housing tenure and the location of the property. In England and Wales, for example, although all owner occupiers and tenants in the private or social sector are eligible for disabled facilities grants (DFGs), local housing authorities and housing associations commonly use their own budgets, such as housing revenue account and housing association funding, to undertake adaptations for their tenants. Therefore, DFG is the main source of funding for private sector adaptations.

According to the “Housing Grants, Construction and Regeneration Act 1996”, to award a DFG, the housing authority must be satisfied that an adaptation is necessary and appropriate to meet the applicant’s needs and that the work is reasonable and practical in terms of the property’s age and condition. Also, the housing department needs to consult the social services department in deciding the necessity and appropriateness of the adaptation work. Therefore, there are at least two departments involved in the adaptation process, with the social services providing assessments and the housing services awarding grants. This multiple organisational arrangement becomes more complicated under two-tier administrations in England, where the county council is responsible for social services and the district council for housing services. Since the Local Government and Housing Act 1989 provided financial support for home improvement agencies (HIAs), many local authorities have worked with them to deliver housing adaptations. With the involvement of multiple organisations and multiple departments, the adaptation system is fragmented and confusing. Clients often have to deal with a network of organisations and numerous professionals when applying funding for housing adaptations [26,27]. Although DFGs are mandatory, they are subject to means test to determine whether an applicant has to make a financial contribution towards the cost of an adaptation. In addition, there is a maximum award limit for DFG, £30,000 in England and £36,000 in Wales. The system requires the applicant to pay any cost that exceeds the statutory maximum.

For a successful adaptation, an applicant has to navigate through a number of procedural steps, including referral, allocation, assessment, funding and installation [28,29]. The adaptation process usually starts when an applicant is referred by their GP or other healthcare professionals to the welfare authority, such as the social services in England and Wales and the social work in Scotland. In some councils, clients are able to refer themselves for adaptation services. On receipt of referrals, an initial screening process normally takes place to prioritise cases and allocate them to specific fieldworkers like occupational therapists (OTs) for assessments. Immediately after allocation of each case, the OT makes the first home visit, assesses the client’s needs against the eligibility criteria and decides the type of adaptation required. The case is then passed on to the housing department for funding authorisation; the grant officer will send the client an application form and associated documents for means testing. Agencies, such as HIAs or care and repair (C&R), are often informed to help clients complete the grant application and the installation work. Before starting any construction work on site, there need to be landlord’s consent when the client is a tenant and planning permission when the adaptation involves an extension or a structural alteration [30]. Once plans and specifications for an adaptation are confirmed, contractors are invited to submit quotations for its installation. A contractor is selected by the client to carry out the work; the finished work is then inspected and approved before payment can be made. These complex procedures have resulted in slow service pathways for housing adaptations and unnecessary waste of limited public resources [31,32].

As the current adaptation process is fragmented and involves several linear steps, the length of time taken to complete these steps decides the efficiency and effectiveness of the whole process. If significant delays occur during this process, older clients may have to move into care homes or be transferred to hospitals. Such a result will cause additional stress and reduced quality of life for the elderly. Previous research [29,31,33] found that long waiting time was a longstanding problem in the delivery of adaptations and had attracted frequent complaints from service users. Recently, national strategies have placed a particular emphasis on tackling delays associated with the adaptation process [34,35,36]. However, there were still frequent reports by service providers and services users on long waiting lists during housing adaptations [33,37].

There have been several studies on evaluating housing adaptation practices. Clayton and Silke [30] evaluated housing adaptation grant scheme with a framework covering effectiveness, consistency, impact and prioritisation. Bibbings et al. [38] reviewed the program of independent living adaptation and identified strengths and weaknesses of the delivery system by measuring quality, speed, appropriateness and value for money. Kempton and Warby [39] measured the social return on investment of adaptations through reduced costs, increased safety and improved well-being and independence of older people. Chiatti and Iwarsson [15] postulated that three aspects had to be considered for an evaluation of home adaptation interventions, including the perspective of the evaluation, the evaluated content of the intervention and the time frame. Time is a common criterion in all these evaluation studies. However, none of these studies reported empirical data of time on the current housing adaptation practice. This study seeks to fill this knowledge gap and to draw international attention to the importance of addressing the housing crisis for older people. It also contributes to the ongoing global debate about creative housing solutions to the big problem caused by the rapidly aging population. This study is aimed at reviewing the timelines of the adaptation process in local authorities across the UK and examining the causes of delays in the current practice. It seeks to address the following questions: 1. What is the average waiting time for the whole process and for each key stage? 2. Where are extensive delays experienced by clients during their application for adaptations? 3. What are the main causes for these delays during the housing adaptation process?

## 2. Research Methods

### 2.1. Sampling and Participants

This study adopts a mixed-methods approach [40,41]. In the first phase, a questionnaire survey was carried out to investigate how local authorities organise their adaptation services. This was followed by five interviews, two specific cases and a focus group meeting with stakeholders to gain different perspectives of the adaptation system. Quantitative analysis is used to examine the current status and identify issues within each stage of the adaptation process in different local authorities, while the qualitative data are used to supplement the qualitative findings. The rationale for the choice of mixed-methods is that neither quantitative method nor qualitative method, on its own, is sufficient to evaluate the effectiveness of housing adaptation practices but the combination of the two can produce a more comprehensive analysis [40,42]. The research chooses to focus on homeowners and private tenants, as they account for the majority of households in the UK and most of them have little knowledge about where to start and what assistances are available when they need adaptations.

Purposive sampling [43] was used for the survey, as local authorities have the powers and duties to provide housing adaptations in the UK. County councils in England were excluded as they do not fund adaptations directly and only provide OT assessments. Also, local authorities in Northern Ireland were excluded, as they have a unique Health and Social Services under a unified structure that leads to organisational differences towards adaptation services from other nations in the UK [37]. As a result, the questionnaire was sent to the remaining 378 local authorities in England, Scotland and Wales.

After the survey, a sample of survey respondents and relevant professionals were approached for semi-structured interviews. This sample was selected purposively, with specific criteria including those currently working in local authorities or associated organisations and being responsible for different stages of the adaptation process. The rationale for this purposive selection is that professionals, who have been involved in the adaptation process, have in-depth knowledge of the existing delivery system. Five professional participants were invited for face to face interviews.

To examine service effectiveness, it is essential to capture service users’ experiences and views of their adaptation services. During the interviews with staff from the adaptation service provider—C&R, two client participants were identified as case studies. The selected clients were older adults (aged 65 or over) with disabilities, living in private properties and had received a housing adaptation grant within the previous two years.

One local council was chosen for a focus group meeting, because it has the social work department, the housing department and C&R working in partnership to provide adaptations and was accessible to the researcher. Participants of the focus group included one OT, one housing surveyor, one grant officer, one technical officer, one C&R manager and one administrative assistant; they have worked together and attended regular meeting for the delivery of adaptations over 2 years.

### 2.2. Data Collection

As explained earlier, the application process for housing adaptations consists of five key steps, including referral, case allocation, assessment, funding approval and installation. The questionnaire asked about waiting time for these five stages and for the whole process. Local authorities were allowed to describe their own processes, where these may be different from the five standard stages in the questionnaire. Once the questionnaire was designed, a pilot test was conducted with 20 local councils prior to the main survey. Results from the pilot led to modification of some questions. The finalised questionnaire, along with a cover letter and postage prepaid envelope, was sent to the housing department of all local authorities across the UK. At the same time, an online survey was activated for those who preferred to respond online. After four weeks, reminder phone calls and emails were made to non-respondents. At the end of this process, a total of 112 local authorities responded to the survey, with 61 sent back by post, 28 replied online, and another 23 by emails. The response rate was 29.6% that is comparable to other studies, such as Connell et al. [44] and Davies et al. [45].

Following the quantitative study, qualitative data were gathered through interviews with five professionals, two case studies and a focus group. The five identified professionals were interviewed in their offices, which lasted between 60 and 150 min. The interview questions were open-ended, focusing on service delivery timelines and blockages within each key stage. Two case studies were carried out with older clients to explore their experiences and concerns of the adaptation process. The dates for every stage of the adaptation process were also collected from both clients. A focus group meeting was organised in one local council; it discussed key issues with local adaptation arrangements and main causes for process delays. All interview and focus group discussions were recorded and transcribed verbatim afterwards.

### 2.3. Data Analysis

This paper provides empirical analysis of waiting times for key stages and causes of delays in adaptation provision that were captured through quantitative and qualitative research. Data analysis was conducted based on the integrated procedures in the mixed-methods approach [40]. Initially descriptive analysis identified the minimum, average and maximum waiting times between process steps across local authorities and measured an overall effectiveness of service delivery. These quantitative results then were discussed with support of qualitative data from the interviews and focus group [42]. For the case studies, the number of days for each client to wait between stages of the adaptation process were calculated. Significant statements on process delays were identified; content analysis was used to identify any causes of delays in the provision.

### 2.4. Ethics

This study complied with the university’s research ethics policy and was approved by the University Research and Ethics Committee. Informed consent was obtained from all participants before interviews, case studies and the focus group.

## 3. Results

### 3.1. Overall Average Waiting Time

Table 1 shows the average waiting times between key stages of the adaptation process. As some local authorities introduced their own stages and some provided partial records, the information is presented under three categories, each of which describes different process patterns and timelines through the adaptation process. Category I includes 35 local councils and shows the timelines across five key stages of the process and the total time, as listed in the questionnaire. Here, the total time is not a simple sum of delays of all stages but a reported length of time taken to complete the whole process. Category II includes 24 local authorities and presents waiting timelines between three stages of provision. These local authorities merged the first three stages from referral to assessment into one, as presented in Table 1 and described by a housing officer:


*1. Referral to OT recommendation is normally 3 months; 2. OT recommendation to grant approval is 60 days; 3. Grant approval to installation is 60 days; 4. The total time is 180 days.*


Category III includes 43 local councils and only displays waiting times for the two stages of funding approval and installation. Due to different departments and agencies being responsible for different stages, some partners, particularly in local authorities under the two-tier system, could only provide their own delivery times. A grant officer reported:


*The referral to assessment stages are not known as they are belonging to the county council. The OT recommendation to grant approval stage is 189 days. From grant approval to installation is 152 days.*


On average, the total length of time for the whole adaptation process in Category I and II was 193 days and 243 days respectively, while the two stages from OT recommendation to installation in Category III took up to 227 days.

The results showed great variations in the application processing speed in different local councils. For example, from the initial request to case allocation, the quickest local authority in Category I took just 1 day while the slowest needed 189 days; the average is 41 days. Following case allocation, the OT required 1 to 103 days to make the first assessment visit; the average time was 21 days. Once the assessment started, it could be expected to complete within a minimum of 2 days and an average of 46 days, but a maximum of 233 days was not exceptional. When the OT assessed the client’s need and specified the required adaptation, the case was passed to the grant officer for funding approval. The time taken to obtain this approval varied markedly across local authorities from 3 days to 233 days, with an average of 85 days. Once a grant was authorised, the installation work could go ahead. It could take 14 to 90 days to finish, with an average time of 54 days. To complete the whole process, the quickest local authority took up to 60 days, while the slowest needed 360 days. Compared with Category I, the timelines in Category II and III showed greater variations within each stage and longer waiting between stages. In Category II, time from referral to assessment varied from a minimum of 28 days to a maximum of 573 days. The time taken in getting funding approved was much longer, with an average of 118 days in Category II and of 112 days in Category III, and could be up to 630 days in Category II and 385 days in Category III. Similarly, the average time taken by the contractor to complete an adaptation was over three months in both Category II and III.

It was common practice that local authorities applied the eligibility framework to prioritise cases and urgent needs were dealt with immediately while other applicants were placed on the waiting list. A social worker confirmed:

*We have different time targets for assessment of different priorities. So if the case is priority 1, the client* will be seen between 24 to 48 h; if it is priority 2, within 72 h; if it is priority 3, must be seen in 28 days; if it is 4, *then it is 12 weeks.*

In addition, simple adaptations where assessment and installation could be done quickly were usually provided straightaway, while more complex needs tended to require more work thus experienced longer waiting times:


*There are huge variations for total time—simple cases can be 3–4 months and complex cases can be several years. (a grant officer)*


### 3.2. Case Studies

#### 3.2.1. Case One

Client A was a woman aged 79, living alone in an upper flat of a block. There was no elevator and she had to manage twelve steps to the entrance door. She had an upper and lower limb weakness that limited her mobility and also had difficulties getting into her bathtub. She was an owner-occupier and had applied for mandatory grants to replace the existing bathtub with a level access shower tray. Client A received her new shower with funding of £3624.32 on 21 January 2015 and the whole process took around 15 months (Figure 1). She was initially referred by C&R to the social work department on 13 October 2013 and 50 days later, the case was allocated to the OT for assessment, which was completed on 30 December 2013. C&R, an adaptation agency, involved after the OT completed the assessment. It provided a range of assistance, including supporting the client to access grant funding and coordinating the installation process. Within two weeks after receiving the case, C&R visited the client on 12 March 2014 to look at the property’s condition, offer technical and architectural advice about the adaptation, and check the client’s entitlement to benefits. Eighteen days later, when the specification and appropriate technical drawings for the adaptation were produced, C&R invited contractors to visit the client with a view of providing quotes for the work. Once an estimate was received, C&R prepared all the relevant documents, including planning permission, building insurance, property deed, relevant certificates and benefits evidence, on behalf of the client for grant application. This took nearly two months from 20 March 2014 to 13 May 2014. Furthermore, the housing department took nearly five months to approve the grant application; the client had to wait for more than three months after grant approval before using the new shower tray.

Client A’s timeline was consistent with the general trend of significant delays during the two stages of funding and installation. The longest wait of 143 days remained at the funding approval stage, which was mainly caused by limited available resources in conjunction with large backlogs of grant applications. Because the client’s property is a typical flat, the adaptation work requires a building warrant. Applying for this warrant took roughly six weeks, resulting in an elapse of 58 days from grant approval to contractor instructed. Installation work took 62 days, within which the client postponed the process for two weeks in order to celebrate Christmas and New Year with her family. Likewise, the client experienced significant waiting times from referral to allocation, although assessment of need was undertaken shortly afterwards. Normally, once the assessment is completed, the OT closes off the case and soon passes it to C&R. However, in this case, it took 59 days for the case to come to C&R after assessment. This substantial delay reflected fragmented responsibilities and the lack of partnership working. In fact, C&R did not know until the case came from OTs and was not able to start the subsequent process as soon as assessment was completed. After C&R took over the case, there was another prolonged wait of 75 days and the main cause was preparation of all supporting documents for grant application, including an application form, an OT report, an approval of pension credit, a schedule of works, two priced estimates, a copy of the title deed.

#### 3.2.2. Case Two

Client B was a man of about 75 years of age, living in a detached house where he and his wife had lived for around twenty years. Since his wife passed away three years earlier, client B had problems with his legs; his mobility deteriorated and went from using a walking stick to sitting in a wheelchair. To get into and out of his home, he used a lift that carried his wheelchair up and down the stairs at the main entrance door. He had applied for grants for the installation of an external ramp in order to exit the house on his own and participate in social activities. Furthermore, as he could no longer manage to get into the bathtub, he had applied for grants for remodelling the bathroom to install a shower unit. Client B was initially referred by his doctor to the social work department on 10 April 2014 and five weeks later, he received a letter from the local authority that acknowledged receipt of his referral. On 26 May 2014, he received another letter concerning a firm date for the OT’s first visit. On 3 June 2014, the OT visited his house. Thereafter, the process seemed to have stalled until 10 July 2015 when the case was passed to C&R. During this period, of more than one year, the client spent time in a hospital twice because of health problems, with the first stay for three weeks and the second for seven weeks. When the case did eventually arrive at C&R, on that very same day a C&R officer visited the client to explain the process, describe the building work and provide information needed for the funding application. When plans and specifications for adaptations were ready, the technical officer invited contractors to the client’s house for tenders. On 19 August 2015 C&R assisted the client with submitting an application form together with all supporting documents for the grant. Funding was granted within two weeks and the selected contractor was then instructed. On 12 December 2015 the case was completed and the whole process took around one year and eight months (Figure 2).

In this case, the longest waiting times occurred at the assessment stage and the installation stage, which accounted for 81.4% of the total time. Client B experienced a significant delay in accessing the OT assessment. There was an unacceptable wait of 402 days, causing by a couple of factors. First, due to deterioration of health condition, the client was taken to hospital twice and stayed for a total of 10 weeks during the assessment process. Each hospital stay had resulted in the assessment being suspended until the OT was informed to resume the work. Secondly, poor arrangement and cooperation between different OTs was reported to delay the assessment process. Client B started the assessment with one OT, who was five months pregnant and off for maternity leave three months later. After that, another OT came out to re-assess the client, which took another several months. Thirdly, the OT worked part-time which caused delays in the assessment process. As an OT just worked for certain days, she was always fully booked in the week and the client always had to wait for another week when he needed to be seen. Besides, a lengthy wait of 96 days was also evident in the installation of the external ramp. The major problem with this was the supplier, who did not provide the ramp within ten weeks. Without the equipment, the contractor could not carry out any installation work. As a result, the case was put on hold and there was a ten-week delay in installing the ramp. Another issue was to obtain a building warrant for installation of the external ramp, leading to delays of up to four weeks. Furthermore, the client complained about the long waiting time when his doctor referred him to the hospital clinic who then made referral to the specialist in the social work.

### 3.3. Main Causes of Delays

#### 3.3.1. Insufficient Resources

The analysis of timelines indicated that it took an average of 6 months to deliver an adaptation and that delays could occur at any stage of the provision chain. In fact, 81.7% of the survey respondents believed that their clients frequently or sometimes experienced delays in the adaptation process. These delays were connected to a range of factors. First, the combination of unanticipated high demand and limited available resources, including finance and staff, was one main cause for delays in providing adaptations:


*Lack of funding had led to DFG cases being held back—at the end of the financial year, this can be up to 8–12 weeks delay before grant being approved. (a grant officer)*



*Not enough staff to deal with delivering adaptations, which means there is a waiting list, approximately 16 months at present. (a housing officer)*


The second case study showed that it was crucial to provide adequate supply of OTs; relying on one part-time OT alone could delay the assessment process. This was also confirmed by a policy officer and a housing officer:


*There are huge variations in the number of occupational therapists per population in each area. The waiting lists for assessment vary considerably.*



*The main issue with adaptations in our area is the time it takes for social services to provide an OT report. If we had more OTs, it would speed up the process considerably and we could then extend help to more people.*


These delays could impact the client’s ability to live independently and increase the need for residential care, leading to a waste of public resources [46,47]. For example, in one case, the installation of a stairlift took 18 months at a cost of £2700. During this time, the applicant used 5 h of additional home care every week costing £3850 in total [48]. The Audit Commission further calculated that one year’s delay in providing an adaptation to a client costed up to £4000 for extra home care [49]. Therefore, it is essential to bring in sufficient resources to tackle delays in the adaptation provision, as suggested by a housing officer and a social worker:


*More money and more staff to deal with adaptations quicker so the client is not waiting as long as they currently are doing.*



*Resources are always the key. When we had more staff, DFG was delivered within the 10 weeks using 4 full-time officers. Current circumstances in the local authority have resulted in a change; procurement is in place and times are around 20 weeks with just 1.7 officers.*


#### 3.3.2. Lack of Joint Work

It was quite common to find that three or more organisations working together to deliver an adaptation job. Ineffective joint work between these partner organisations at different stages was reported as a potential hazard to the successful service delivery. An OT pointed out:


*Sometimes we didn’t get enough information from social workers. Delays generally occur during peaks in the number of referrals.*


Such complaints became more frequent when the adaptation process spread over more than one local authorities, as reported by a housing officer:


*OT in the county council has to shut their cases down once it is passed to the district council. If the case has any question or changes, we have to request to reopen the case again in the county council. It takes time as it could be a different OT to deal with the case.*


This disconnection was also found in the second case study, in which the OT did not pass the case to C&R soon after assessment, resulting in the client had to wait for nearly two months. This was confirmed by a HIA officer:


*We know until the cases come to us from OTs. Apart from that, we don’t have details or would not be told what we will receive. So we don’t know who has been assessed and is on the waiting list, we just do what tell us.*


It is clear that the lack of partnership working can lead to unnecessary delays in delivering housing adaptations. In addition, poor communication and ineffective arrangement can withhold the process. In the second case, because the second OT did not collaborate with the first OT before she took maternity leave, Client B had two assessments that took a number of months and resulted in delays of over one year during the assessment stage.

#### 3.3.3. Bureaucratic Procedures

Some delays were the symptom of bureaucratic procedures and excessive paperwork, such as the landlord’s consent and planning permission. Any kind of adaptation in the private rented sector, no matter how minor, could not be carried out until receiving the landlord’s permission. The process was described by administrative staff as lengthy but necessary:


*We have to contact the landlords and get their approvals for the building work. Some landlords are very fast and good, the clients can get the letter within a week. But in some cases, it probably takes two or three weeks.*


In the case of structural changes to a property, planning permission must be obtained before an adaptation could go ahead. These additional procedures are more time-consuming and expensive, as commented by a C&R officer:


*When a client stays in hospital, it costs around £4000 a week. If the client needs a shower adaptation and we can get it done in two working weeks, the adaptation will cost roughly £3500 and the hospital stay will cost £8000. But if we follow the legal procedures, it could be 6, 8, 10, 12 weeks. If we say 10 weeks, that is £40,000, minimum.*


This was also evidenced in both case studies; Client A waited for approximately six weeks to receive the building warrant while Client B for four weeks. Likewise, as these bureaucratic procedures take place during funding approval and installation, the survey found that the two stages were where the main blockages occurred in the adaptation process, with the longest waiting time of 139 days in Category I, 211 days in Category II and 227 days in Category III. Because these are legal requirements, housing officers felt completely helpless about the time taken for them:


*I find that the legal requirement for planning consent is annoying as it extends the process by several weeks.*


In addition, these legal procedures placed a particular restriction to what can be achieved as they must be adhered to, as reported by a housing officer:


*There is a reluctance to move away from the legal framework of the mandatory grant. The attendant bureaucracy adds delays. Attempts to develop “fast track” schemes fail due to fears that non-mandatory grant scheme will fail to get funding via Communities and Local Government.*


Therefore, there is a need for central government to review the legal framework governing the provision of grants and empower local authorities to be more flexible in carrying out housing adaptations. For example, service providers may be allowed to carry out adaptations first for urgent cases and to complete the paperwork retrospectively, as suggested by a C&R manager:


*The hospital will discharge the client when we get the adaptation done. If we can get the adaptation done and then follow up with the paperwork, the client can get out of hospital more quickly. Not for every single one, only for the high priority cases, the big ones.*


#### 3.3.4. Gap between Grant and Cost

The need for applicants to make extra financial contributions to cover the difference between the amount of grants they received from local authorities and the total cost of their adaptations can also delay the process from funding to installation [30]. Normally mandatory grants (e.g., DFGs) are issued subject to a means test and a maximum grant limit—that is, clients, who have certain earning/saving, or need adaptations costing above the upper limit, would be asked to pay for any works in excess of their grants. When clients have any difficulty to make their contribution towards the whole and part of the cost, the funding process would break down and a delay occurs. The survey showed that the longest average waiting time was at the funding approval stage. A housing officer and a grant officer also pointed out:


*There are more difficulties when clients have to pay for the 20% of the cost; they have to find charitable funding to assist at times, resulting in delays to the delivery of adaptation works.*



*So the time taken to find the top-up funding needs a couple of months. This will delay the process.*


A delay in raising the additional funding might trigger a second delay caused by price inflation or might lead to early closure of the case without any adaptation being done. As noted by a local authority in Perry’s study, approximately 25% of the cases, with DFGs granted, did not complete due to the lack of client’s contribution [47].

#### 3.3.5. Shortage of Reliable Contractors

The survey revealed that the two stages of funding and installation were major barriers to efficiency of the adaptation provision. The major source of the longest waiting time, between funding approval and installation, was reaching the agreement of the schedule of work and getting contractors engaged and then finishing the job. Sometimes, it was unacceptably slow to finalise the specifications of an adaptation and to secure a contractor for carrying out the work, as complained by a grant officer:


*The process takes for too long. The process would be easier and more efficient if we can build up the speed in doing drawings, putting on to tender and starting the installation.*


In some local areas, finding a reliable contractor was difficult and time-consuming:


*One of the main delays in the system is the lack of availability of builders to do the work. (a housing officer)*


To help clients get the requisite number of quotes in a timely manner, 70.9% local authorities kept an approved list of contractors. Those, who have not yet compiled such a list, should be encouraged to produce it in order for clients to identify suitable contractors quicker and speed up the installation process:


*Maximise availability of contractors to undertake works, therefore minimising waiting time for work to start on site. (a housing officer)*


Delays were frequently seen after the contractor had started the adaptation work on site. They might be caused by any of the following: lack of materials/equipment, shortage of skilled laborers, or delay of interim payments [30,50]. For example, in the second case study, the installation process was delayed for ten weeks because the contractor’s supplier failed to provide the external ramp. This isolated the client from any social or community activities and left him at risk of injury or harm, as complained by the older person:


*I was unhappy with this supply, it took 10 weeks and should be here earlier. That made me worse, I cannot go outside to see my family and visit my friends.*


Often clients tended to choose the lowest tenders, who may not be the right contractor for a speedy completion. A C&R officer said:


*Normally there are two quotes, the grant is based on the lowest quote and most clients go for the lowest one. The lowest quote, unfortunately, takes the longest time to get this done.*


#### 3.3.6. Clients’ Decisions

Another major delay factor was the client’s own decisions. In principle, the decision about when to start the building work remained in the hands of the client. The process was often delayed for weeks, or even months, when a client took control of the progress, as highlighted by a housing officer and a social worker:


*The council operates an application process which affords the applicant with as much control as possible over the destiny of their application. While this can at times present delays in the system it leaves maximum control with the client.*



*You will be surprised by a number of people who just take for ever to get the ramp or shower done. You would think that they will do it straight away, but they don’t. Why? No idea.*


Even worse, after holding up the process for a long period of time, some clients might decide not to go ahead with their adaptation, as pointed out by a C&R officer:


*The clients have the choice, they can turn around and say I don’t want it, even the grant is approved and everything is ready.*


Some clients turned down the council or agency’s help and spent more time to appoint their own contractor, as reported by a housing officer:


*There can be a time delay between assessments and the work being carried out due to the selection and appointment of contractors to carry out work which owners are involved in.*


In addition, there might be further delays when the client’s own chosen contractor lacks relevant skills and experience to undertake the adaptation work:


*When applicants arrange their own works, we have, on occasion, concerns about the quality of the work of their contractors. They tend to take longer than us to organise from start to finish. (a housing officer)*


## 4. Discussions

Waiting time has been identified as a key benchmark against which effectiveness of adaptation provision is assessed [29]. In practice, most local authorities record the timelines across key stages of the adaptation process. Their records, however, varied substantially. The whole process is broken down by different authorities into different stages. This made comparison in terms of service efficiency across local councils difficult. Without such a comparison, it is almost impossible to benchmark service performance and to produce a more efficient system. Therefore, there should be a uniform procedure to record delivery times for all the steps of the adaptation process across all local authorities. More accurate time recording can help to improve performance. This is also found by this survey, local councils who recorded waiting timelines for all five stages, those in Category I, delivered quicker adaptation services than those who did not, councils in Category II and III.

Although almost all local authorities responded to the timelines question, some had provided partially completed answers. The area with the most omissions was details from referral to assessment, as information on these stages was held by different departments within one local council or across different councils. This showed one of the flaws of the current system that partner organisations within one authority or across different authorities did not have a shared database for adaptation provision. On the other hand, strong links between different stages of the provision chain are the key to an efficient and seamless service [26]. If one partner is unable to access all the necessary data from other partners within a local authority or from another authority, as was the case for some of the survey respondents, it is difficult to capture what is happening in the adaptation delivery system and how to improve the service performance [27]. The joint working arrangement has a significant impact on the waiting time between the process stages. Therefore, it is essential for local authorities to develop a shared system and a high standard of coordination, which will enable all the partners to process cases quickly and to minimise the negative impact of the fragmented service delivery.

All stages form an integral part of the sequential adaptation process; each stage must be completed before the next can begin [50]. When a blockage occurs at one stage, the whole process breaks down and the client cannot receive their adaptation in a timely manner [28,51]. To improve the efficiency of the adaptation process, it is necessary to identify the existing weak links in the provision chain, or the stages where the waiting time is long and delays are frequent [29]. According to the survey, the average waiting time for each stage was quite high; delays were frequently found at different stages. The longest waiting time was found during the funding stage and the installation stage, accounting for over half of the total waiting days. On the other hand, the lowest level of waiting occurred at the case allocation stage and assessment was also undertaken within a relatively short space of time. This indicates that the assessment process, which traditionally had longer waiting time [31,46], has been streamlined considerably. Finally, the wait time from the initial request to case allocation was comparatively high. 

Overall, the time taken to complete an adaptation was still unacceptably long; clients had to wait for months, or even years, before their adaptations were delivered. To reduce the waiting time and ensure a smooth process flow, waiting time targets for each key stage of the process from referral to completion had been proposed by Clayton and Silke [30] and Audit Scotland [52]. This proposal had set off a heated debate, which was also reflected in the survey. Some officers believed that setting timescale for the adaptation process and its stages was the most effective way of avoiding delays and waiting times. However, others argued that setting timescales might speed up the process but bring down the service quality. Such an argument would not be valid if the time for completing each key stage is set reasonably and realistically. In fact, a clear timeline for the process enables all partners to schedule their tasks and complete them in time. The survey also showed that councils in Category I who recorded the time between each key stage completed the adaptation process much faster than other councils in Category II and III. Therefore, introducing a reasonable waiting time for each stage is important to produce an effective process and to minimise waiting time.

The survey showed that there were significant variations in the waiting time not only within each stage but also between stages across local authorities. Within one local authority, there are equally big variations between different cases and adaptation types. Such variability was experienced by the older clients of the two cases studies, even though they were within the same local authority and had their adaptations within approximately the same period of time (i.e., 2013 to 2015). The major blockage in the first case occurred at the stage of funding approval; while in the second case the main delay was at the assessment stage. Although C&R helped both clients with preparing all the information, applying for grants and supervising the building work, there were still significant delays between funding and installation. Notably, because of some factors, including the client’s ten weeks of hospital stay, lack of collaborative work between the two OTs, and the OT working on a part-time basis, the second client had to wait for more than one year for his assessment to be completed. This was an incredibly long wait, indicating that the effectiveness of OTs, although improved to some extent, still needed to be strengthened. Similar to the survey results, the social work department took more than one month to allocate the two cases for assessments, leading to overall delay. Despite these variations, an analysis of timelines showed the patterns of waiting times across the adaptation process and weak links between the stages from initial referral to work completion. Improvement is needed in all main aspects of housing adaptation, including referral management, joint work, funding authorisation and installation process, in order to provide a more effective and efficient service.

There is a wide variety of reasons for delays in delivering adaptations. A common root cause stemmed from a lack of resources, as not enough money and staff were allocated to keep up with the increased demand, as found in this study as well as other research, such as Bibbings et al. [38], Jones [31], Mackintosh and Leather [32]. Additional resources are required to strengthen the capacity of local authorities to provide adaptations. As the adaptation process is administered by multiple departments and organisations, disconnections between them often create unnecessary delays. To achieve a seamless service process, it is important to ensure that all partners within a local authority or across different authorities have real collaborative working. If an adaptation is required for a rented property or it affects a building’s structure, it is necessary to get landlord permission or planning permission. Also, due to the mandatory nature of adaptation grants, there has to be a means test for all applications, except for those from children or young people. Going through these inevitable legal procedures often results in an extended period of time. There is a need to review the legal framework governing the provision of grants and allow local authorities more flexibility in carrying out housing adaptations. Importantly, when funding is authorised, there are still delays while decisions are made by the client on how and when to carry out the work. Delays are more common when clients are directly involved in the selection and appointment of a contractor, as they are often inexperienced to select competent contractors. It requires a clear guidance to make effective involvement of service users who could be in good control of their adaptation.

Although this study contributes to a better understanding of waiting times and causes of delays across the key stages of the adaptation process, there are some limitations. First, it proved to be a real challenge in recruiting all local authorities to the survey and obtaining a high response rate, as staff responsible for providing adaptations always face overwhelming caseloads and participation in research is not their priority. Therefore, data were only collected from less than a third of the councils; and caution is required when generalising the research findings. Secondly, although the researchers adopted measures to maximise validity and reliability of the interview data, there are always potential biases on the interpretation of open questions and the discussion of some aspects. Finally, due to time constraints, the numbers of interviews and case studies were relatively small. Further investigation should be undertaken to get a more complete picture of the adaptation practice in the UK.

## 5. Conclusions

This study investigated the timelines for the delivery of housing adaptations in local authorities across the UK. Overall, the average waiting time for the whole adaptation process from referral to completion was unacceptably long, with significant delays occurring at all stages, especially during funding approval and installation. There was a lack of consistency with wide variations in waiting times for different stages both within a local authority and across different authorities. Main factors of delays include limited resources, ineffective partnership, bureaucratic procedures, funding gap between grant and cost, lack of skilled contractors and the client’s decisions. Moving forward, extra resources including funding and staff are needed to meet the rising demand for adaptations from an aging population. Close coordination between all partners and a clear timeline for each stage are also needed to ensure a quicker and responsive service. It is also important to review the legal framework and its bureaucratic procedures; local authorities should have more flexibility in preparing all paperwork and delivering adaptation services.

## Figures and Tables

**Figure 1 ijerph-16-00192-f001:**
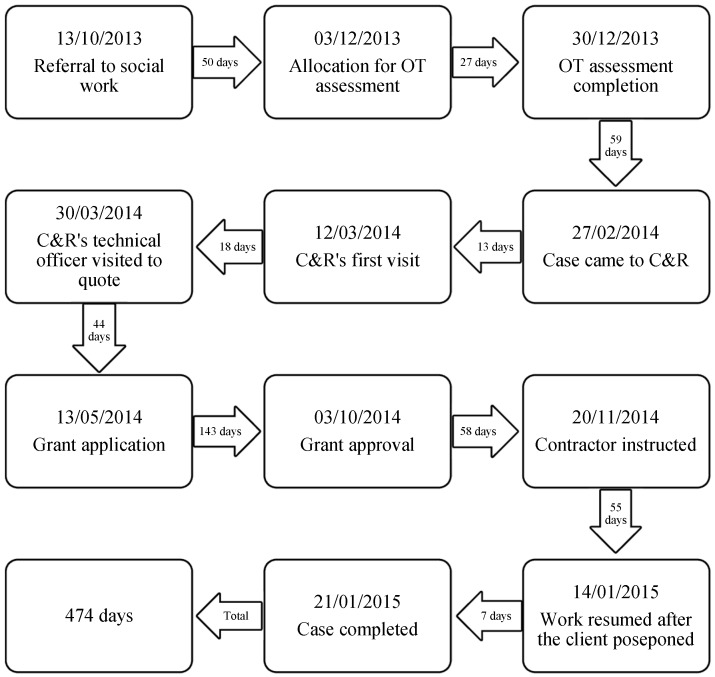
The timeline of the adaptation process for Client A.

**Figure 2 ijerph-16-00192-f002:**
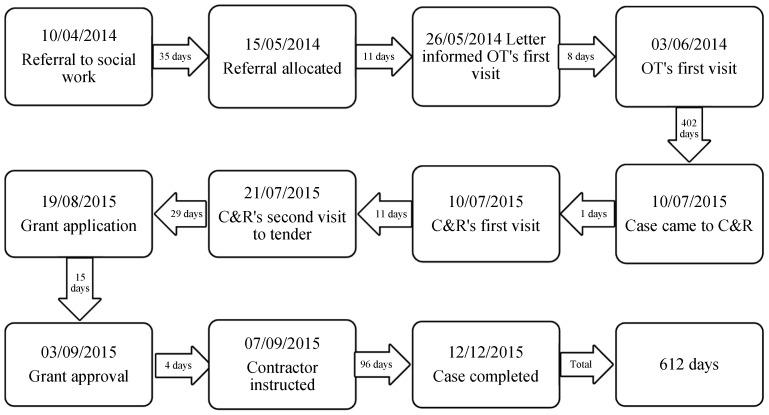
The timeline of the adaptation process for Client B.

**Table 1 ijerph-16-00192-t001:** Timelines between stages of the adaptation process.

The Adaptation Process		Minimum	Average	Maximum	Median
Stages		(day)			
Category I					
1	Referral	1	41	189	28
2	Case allocation	1	21	103	7
3	Assessment	2	46	233	21
4	Funding approval	3	85	233	60
5	Installation	14	54	90	56
	Total	60	193	360	166
Category II					
1–3	Referral to assessment	28	121	573	85
4	Funding approval	23	118	630	67
5	Installation	30	93	226	77
	Total	90	243	474	236
Category III					
4	Funding approval	7	112	385	92
5	Installation	7	115	356	96
	Total	84	227	522	188

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
