# Peer review of "Causes of Delays during Housing Adaptation for Healthy Aging in the UK"

_ijerph, 2019, doi:10.3390/ijerph16020192_

Round 1
Reviewer 1 Report
Reviewers report: causes of delays during housing adaptation for healthy aging in the UK
This study aimed to examine the delays in the arrangement of housing adaptation for older people in a mixed methods design
Abstract: The sentence beginning “these steps” needs a little revision, otherwise the abstract is a fair and concise summary of the study
Introduction:
If the paper is meant for international consumption, then local council might be too parochial a term – perhaps municipal authority (local and regional)?
Otherwise the introduction is well-written and the purpose of the study well articulated.
Methods
The mixed methods approach is appropriate, was the sample based upon any methodological approach – was saturation of themes reached with n=5? Was this the intended methodological approach?
I am concerned that the focus group participants may have been atypical – given the researcher note that this team appears to work together – when the theme overall appears to be one of disconnect in working in partnership.
Results – although clear – I didn’t get the definition of categories I,2,3 – could this be made plain?
It seems that different authorities operated vastly different urgency and triage procedures, depending upon the authority
The section is well written and clear, the organisation is appropriate – the tables are easy to read
Discussion – again well written – the authors should address the potential limitations of their study and address potential biases in terms of response, the limited qualitative data and limits to external validity
Author Response
Response to Reviewer 1 Comments
Point 1: This study aimed to examine the delays in the arrangement of housing adaptation for older people in a mixed methods design.
Response 1: No response required. We thank you for the reviewer’s valuable comments and suggestions on our manuscript.
Point 2: Abstract: The sentence beginning “these steps” needs a little revision, otherwise the abstract is a fair and concise summary of the study.
Response 2: We have followed the reviewer’s suggestion to change “these steps” into “the five stages” in the abstract section. Page 1, line 17.
Point 3: Introduction – If the paper is meant for international consumption, then local council might be too parochial a term – perhaps municipal authority (local and regional)? Otherwise the introduction is well-written and the purpose of the study well articulated.
Response 3: In the UK, devolution settlements transferred responsibilities for housing, health and social care to the newly created Scottish Parliament and Welsh Assembly, the policy for provision of housing adaptations has been determined and operated by the devolved governments. Following decentralization and localism within each nation, the central government determines the “top-down” policy directions and outcomes, while local authorities adopt the “bottom-up” approach to make decisions on implementing policies and action plans.
Regionalisation was rejected on the grounds that setting up regional assemblies not only was an expensive operation but also created an unnecessary bureaucracy. Except for Greater London which has a directly elected London Assembly, other regions in England have become powerless bodies without any territorial administrative function and are now mainly used for statistical and economic analysis. In Scotland and Wales, there are only unitary, single tier council known as local authority. Therefore, local authority is used as a term in the paper as they play a central role in setting out proposals for the provision of housing adaptations and deciding the implementation of these proposals.
Points 4: Methods – The mixed methods approach is appropriate, was the sample based upon any methodological approach – was saturation of themes reached with n=5? Was this the intended methodological approach?
Response 4: This study employs a mixed-methods sequential explanatory strategy in which the quantitative and qualitative phases come in two consecutive periods. In the first quantitative phase, a questionnaire survey focuses on how local authorities plan, organise and monitor their adaptation services. In the second qualitative phase, eleven interviews, one focus group and two case studies with stakeholders explore different perspectives on the statistical results in more depth. In such a research design, the quantitative results inform the qualitative data collection and the qualitative analysis offers in depth explanations for issues identified in the quantitative phase.
Considering the main aim of this study, priority was given to the quantitative method, as it provides the main evidence on the current status and the factors that affected the service process in different local authorities. The lesser qualitative component was targeted at capturing experiences and views of stakeholder representatives on key issues identified by the quantitative results. In other words, the qualitative data is to supplement the quantitative finding. To achieve data saturation, when the questionnaire survey found issues within each stage of the adaptation process, five professional participants responsible for different stages were approached for interviews to identify causes of these issues. This sample was selected purposively – they worked in different departments or organisations and were experienced in the adaptation system. Besides, case studies with two clients were conducted to explore issues with the adaptation system from the service user’s perspective. After interviewing five professionals individually and case studies with two clients, one focus group is organised to elicit a number of perspectives on housing adaptation practices. We have added a brief explanation in the Research Methods section, page 3, line 31-36.
Point 5: I am concerned that the focus group participants may have been atypical – given the researcher note that this team appears to work together – when the theme overall appears to be one of disconnect in working in partnership.
Response 5: Because of the complicated web of legislation, the adaptation process is normally administered by multiple departments and organisations. It is quite common to find that different partners work together to deliver an adaptation job but their partnership arrangement has been poor or ineffective. The focus group was organised in one local council because the council has typically social work, the housing department and care and repair working in partnership to carry out adaptations and participants, including one OT, one housing surveyor, one grant officer, one technical officer, one C&R manager and one administrative assistant, are responsible for different stages of the whole adaptation process. Using this group will enable to measure the level of disconnection and the problems with partnership even the team has worked together for years.
Point 6: Results – although clear – I didn’t get the definition of categories I,2,3 – could this be made plain?
Response 6: As suggested by the reviewer, we have made some changes to clarify the definition of categories 1,2,3 in the Result section. Page 5-6, line 212-223.
Point 7: It seems that different authorities operated vastly different urgency and triage procedures, depending upon the authority.
Response 7: Yes, as there is no legislation or policy that identifies one primary organisation responsible for the delivery of adaptation, different local authorities are allowed to decide on their own guidelines, procedures and eligibility criteria.
Point 8: The section is well written and clear, the organization is appropriate – the tables are easy to read.
Response 8: No response required. Many thanks for the reviewer’s positive feedback.
Point 9: Discussion – again well written – the authors should address the potential limitations of their study and address potential biases in terms of response, the limited qualitative data and limits to external validity
Response 9: Followed the reviewer’s suggestion, we have added the potential limitations of our study and addressed potential biases at the end of the Discussion section. Page 14, line 587-596.

Reviewer 2 Report
Thank you for the opportunity to review this important study on housing adaptation with a focus on the adaptation process. I believe the topic and findings of this study has much potential to contribute to the field. I provide some critical areas for improvement, in my view.
The authors did a very good job of succinct discussion on the importance of housing adaptation and related theoretical perspective (p.1 line 40- p.2, line 51). However, since this study is about investigating the adaptation process of the specific program, it is strongly recommended that the authors bring in more directly relevant conceptual framework that talks about how to evaluate the efficiency and effectiveness of the process or implementation of any program or policy. It was clear throughout the manuscript that the authors' main point was 'timelines' in the process, but this key idea or other major domains were not introduced or discussed at all in the intro/body part. Although necessary, the most part of the intro was spent on presenting background information (line 52 p. 2 - line 101 p.3). As such, a reader was left with the impression that this manuscript was written for project report, rather than a research article. The discussion was a summary or repetition of the findings and little discussion was provided to what extent the findings extend current knowledge about this field. If authors believe this is the first of its kind in evaluating the process of home adaptation, then that point should be made very clear from the relevant literature review.
Author Response
Response to Reviewer 2 Comments
Point 1: Thank you for the opportunity to review this important study on housing adaptation with a focus on the adaptation process. I believe the topic and findings of this study has much potential to contribute to the field. I provide some critical areas for improvement, in my view.
Response 1: No response required. We thank you for the reviewer’s valuable comments and suggestions on our manuscript.
Point 2: The authors did a very good job of succinct discussion on the importance of housing adaptation and related theoretical perspective (p.1 line 40- p.2, line 51). However, since this study is about investigating the adaptation process of the specific program, it is strongly recommended that the authors bring in more directly relevant conceptual framework that talks about how to evaluate the efficiency and effectiveness of the process or implementation of any program or policy.
Response 2: As suggested by the reviewer, we have brought in some discussions on relevant conceptual frameworks that evaluate different sections of housing adaptation programs and provided an explanation of our own conceptual framework for assessment of the adaptation process in the Introduction section. Page 3, line 110-120.
Point 3: It was clear throughout the manuscript that the authors' main point was 'timelines' in the process, but this key idea or other major domains were not introduced or discussed at all in the intro/body part.
Response 3: Followed the reviewer’s suggestion, we have added some explanation on waiting timelines of the adaptation process in the Introduction section. Page 3, line 101-109. Under the Results section, the subsection 3.1 shows the survey results about average waiting time between key stages of the adaptation provision. Based on these results, detailed discussions are given in paragraph 1 to 5 of the Discussion section, page 12, line 499-567.
Point 4: Although necessary, the most part of the intro was spent on presenting background information (line 52 p. 2 - line 101 p.3). As such, a reader was left with the impression that this manuscript was written for project report, rather than a research article.
Response 4: In order to reach an international audience and provide better understanding of the particular context for housing adaptations in the UK, we spent paragraph 3-4 of the Introduction section (page 2, line 54-81) in explaining the legislative framework that sets out the powers and duties of local authorities to provide adaptations. Given the complicated web of legislation, an applicant has to navigate through a number of procedural steps to receive a successful adaptation. So we gave a brief explanation of the delivery process and of service organisations involved in different stages of the process in the paragraph 5 (page 2, line 82-100), which is crucial for the following presentation and discussion of research findings.
We have added background information about waiting timelines across stages of the adaptation process and relevant conceptual framework that assess the efficiency and effectiveness of the adaptation program in the Introduction section (page 3, line 101-109) to improve this research article.
Point 5: The discussion was a summary or repetition of the findings and little discussion was provided to what extent the findings extend current knowledge about this field. If authors believe this is the first of its kind in evaluating the process of home adaptation, then that point should be made very clear from the relevant literature review.
Response 5: We have provided an explanation of our conceptual framework for evaluation of housing adaptation practices in accordance with the reviewer’s comments in the Introduction section (page 3, line 110-120). In this paper, we have the Results section separate from the Discussion section. The Results section only presented relevant quantitative and qualitative results and did not include any interpretation of these results, while the Discussion section mainly discussed all of the results, outlined new findings compared to current knowledge, and highlighted research reflects.
